# Microporous Activated Carbon from *Pisum sativum* Pods Using Various Activation Methods and Tested for Adsorption of Acid Orange 7 Dye from Water

**DOI:** 10.3390/molecules27154840

**Published:** 2022-07-28

**Authors:** Mohamed A. El-Nemr, Ahmed El Nemr, Mohamed A. Hassaan, Safaa Ragab, Luigi Tedone, Giuseppe De Mastro, Antonio Pantaleo

**Affiliations:** 1Department of Chemical Engineering, Faculty of Engineering, Minia University, Minia 61519, Egypt; mohamedelnemr1992@yahoo.com; 2National Institute of Oceanography and Fisheries (NIOF), Marine Pollution Lab, Alexandria 21556, Egypt; mhss95@mail.com (M.A.H.); safaa_ragab65@yahoo.com (S.R.); 3Department of Agriculture and Environmental Sciences, Bari University, 70121 Bari, Italy; luigi.tedone@uniba.it (L.T.); giuseppe.demastro@uniba.it (G.D.M.); antonio.pantaleo@uniba.it (A.P.)

**Keywords:** microporous, activated carbon, pea pods, *Pisum sativum*, Acid Orange 7 dye, pyrolysis, removal

## Abstract

**Highlights:**

**Abstract:**

This work demonstrates the preparation of high-surface-area activated carbon (AC) from *Pisum sativum* pods using ZnCl_2_ and KOH as activating agents. The influence of CO_2_ and N_2_ gases during the carbonization process on the porosity of AC were studied. The highest specific surface area of AC was estimated at 1300 to 1500 m^2^/g, which presented characteristics of microporous materials. SEM micrographs revealed that chemical activation using an impregnation reagent ZnCl_2_ increases the porosity of the AC, which in turn leads to an increase in the surface area, and the SEM image showed that particle size diameter ranged between 48.88 and 69.95 nm. The performance of prepared AC for adsorption of Acid Orange 7 (AO7) dye was tested. The results showed that the adsorption percentage by AC (2.5 g/L) was equal to 94.76% after just 15 min, and the percentage of removal increased to be ~100% after 60 min. The maximum adsorption capacity was 473.93 mg g^−1^. A Langmuir model (LM) shows the best-fitted equilibrium isotherm, and the kinetic data fitted better to the pseudo-second-order and Film diffusion models. The removal of AO7 dye using AC from *Pisum sativum* pods was optimized using a response factor model (RSM), and the results were reported.

## 1. Introduction

Rapid urbanization and the expansion of factories in varied locations pose a huge challenge to humankind in this century in the form of water pollution issues. As a result, treating aquatic pollutants is frequently a difficult task. This is due to the large number of structurally diverse pollutants that can interact with a variety of natural and anthropogenic sources, dissolved or particulate chemical species, light, and even living organisms, causing significant ecosystem disruption [1,2,3]. As a result of rapid urbanization and the proliferation of factories in various environments, water pollution is one of the most dangerous threats we face in this era [4]. Because water pollution has an impact on all living species and populations as well as on the overall performance of the aquatic ecosystem, the fact that dyes are difficult to remove from the water system, stable, and non-biodegradable makes them dangerous water pollutants for the most part [5,6]. The dying, textile, leather, printing, cosmetics, food processing, and paper-making sectors all produce dye-contaminated wastewater. The harmful and carcinogenic consequences of these colored effluents’ discharge onto living organisms make them a serious environmental issue [7,8]. Globally, roughly 7 × 10^5^ tons of dyes (seven distinct types of dyes) are produced each year, and about 10–15% of the dye used in the process is lost in the discharged effluent [9,10]. Due to the hazardous and carcinogenic nature of these effluents, they should never be released into the environment. It is possible to classify dyes into anionic (acid, direct, and reactive dyes) and non-ionic (vat dyes and disperse dyes) categories based on their solubility in water [11,12,13]. Basic dyes can be used to color acrylic fibers, acid dyes to color wool, silk, or nylon, and disperse dyes to color polyester or nylon with. Cotton fibers should only be dyed using vat or reactive dyes. Dye hydrolysis creates intensely colored wastewater effluents that are contaminated. When using vat dyes, it is necessary to reduce them in an alkaline medium with potent reduction agents, such as sodium hydrosulfite, before they can cling to the textile substrate [14]. The majority of dyeing methods employ a variety of pollutant-generating additives. Water dyes not only reduce sunlight penetration and thus photosynthesis, but they may also impede aquatic biota growth and reduce the solubility of gases in water [15,16]. Conventional wastewater treatment systems have difficulty removing many colors from water because they are nonbiodegradable and persistent under light, heat, and some oxidizing agents [17]. Efficient dye removal from aqueous solution by high-performance electrospun nanofibrous membranes through incorporation of SiO_2_ nanoparticles has been reported [18]. Activated carbon is a porous structure material showing amphoteric characteristics. Therefore, it is used for adsorption of inorganic and organic compounds such as dyes. There are some notable advantages of activated carbon (AC), such as low production cost, high surface reactivity, significant thermal stability, active free valances, and the applicability of its high capacity and high rate of adsorption [19]. Both powdered (approximately 44 µm) and granulated (particle size 0.6–4.0 mm) forms of activated carbon can be obtained [20,21]. Activated carbon can be divided depending on its pore size into microporous (<2 nm), mesoporous (2–50 nm), and macroporous (>50 nm) groups, which can be controlled during the preparation process of activated carbon for specific applications [22,23]. There are two primary methods for activating carbon compounds: physical and chemical. Chemical activation uses higher temperatures, which make physical activation more efficient. As a result, improvements in activated carbon’s porous structure have been made when using a chemical activation approach [20,24]. The physical activation process is performed in two steps. The first step is carbonization to produce non-porous char. The second step is the activation with carbon dioxide (CO_2_), N_2_, or the flow of steam [25]. Chemical activation can be performed in one step by applying raw material thermal decomposition with chemical reagents. Chemical reagents used in the activation processes are either acidic reagents, such as zinc chloride (ZnCl_2_) [13,14], phosphoric acid (H_3_PO_4_) [26], hydrochloric acid (HCl) [27], and sulfuric acid (H_2_SO_4_) [28] or basic reagents such as potassium hydroxide (KOH) [29], potassium carbonate (K_2_CO_3_) [30], sodium hydroxide (NaOH) [31], and sodium carbonate (Na_2_CO_3_) [32]. Acid-based activation processes lead to the formation of -COO groups on the surface of activated carbon in addition to increasing its surface area [32]. The negatively charged carboxylic groups increase the ability of carbon to adsorb heavy metals, such as lead, cadmium, cobalt, and mercury. Additionally, acid activation leads to a decrease in ash content in activated carbon and enhanced activated carbon properties [33].

Activated carbon’s price continues to rise due to the increased use of this material in a variety of processes, including adsorption, fuel cells, gas storage, and catalysis, to name a few. Therefore, the search for new, easily available, and cheap resources for the production of activated carbon is an important goal. Due to their availability, renewable nature, and ability to reduce environmental pollution, agricultural leftovers have received considerable interest as raw materials for synthesizing activated carbon [34]. Therefore, many agricultural-plant-based materials, such as cotton hull, sago industry waste, banana pith [35], sawdust [36,37,38], fiber [39], shells from different sources [40,41,42], palm [43], date pit [44], rice bran [45], coir pith [46], mango seed [47], straw [48], coconut coir dust [49], pine cone [50], wheat husk [51], pomegranate husk [28], banana trunk waste [52], and spent coffee grounds [53] have been used for the production of AC for wastewater treatment. Due to the multiple uses of AC, many researchers attempted different methods to reduce the cost of activated carbon production by generating carbon from cheap available sources or by performing surface modification. However, few research works have been reported for the use of pea skin for production of activated carbon [54].

The use of *Pisum sativum* pea pods for the preparation of microporous activated carbon using different chemical activations under flow of N_2_ and CO_2_ was investigated. The prepared activated carbon with the highest surface area was used to study the uptake of Acid Orange 7 (AO7) dye from artificial wastewater. AO7 dye removal was investigated as a function of several variables, including temperature, initial dye concentration, adsorbent concentration, and contact time. Different isotherm and kinetic models were used to calculate and explain the adsorption isotherms and kinetic parameters, which were then explored in detail.

## 2. Results and Discussion

### 2.1. Effect of Impregnation Reagent, Activation Temperature, and Activating Reagent

For the preparation of activated carbon, different processes were performed by chemical activation with a 2:1 ratio of ZnCl_2_ and with 10% KOH. Additionally, the carbonaceous precursor was activated by carbon dioxide and nitrogen gases under different temperatures (600, 700, and 800 °C) for 1 h. These impregnation reagents and their ratio have been chosen according to our previous study. According to the experimental results, the type of impregnation reagent is the critical parameter among the operational parameters of the activation reaction in order to prepare *Pisum sativum* pods as activated carbon (AC), as it has a significant effect on the specific surface area, monolayer volume, mean pore diameter, total pore volume, and adsorption capacity of the activated carbon.

We investigated the effect of activation temperature on the AC yield. The yield of activated carbon reduced from 36.11 to 20.94% and from 37.61 to 18.89%, respectively, when 10% KOH was activated with N_2_ and CO_2_ gases at activation temperatures of 600 to 800 °C. From the previous results, we deduced that the yield of activated carbon decreased because the KOH may have accelerated the evaporation of volatile compounds through the breakdown of aromatic and aliphatic chains. However, a slight decrease in yield from 43.61 to 31.58% was observed by using a 2:1 ratio of ZnCl_2_ under CO_2_ gas at activation temperatures (600–800 °C). On the other hand, under N_2_ gas, differences in the activated carbon yield at the activation temperatures 600 to 800 °C were not observed. Generally, the yield of AC-KOH is smaller than other AC-ZnCl_2_ samples. The reason for this could be that the zinc chloride acts as a dehydrating agent which enhanced the removal of volatiles such as H_2_ and O_2_ from the carbon matrix.

Chemical activation was used to evaluate the effects of KOH and ZnCl_2_ on the carbonization of activated carbon from *Pisum sativum* pods. Table 1 summarizes the preliminary findings. The results indicated that utilizing impregnation agents ZnCl_2_ and 10% KOH for chemical activation, as well as activating agents N_2_ and CO_2_ gases at activation temperatures of 600, 700, and 800 °C, can result in considerable changes in the surface area and pore volume of the final activated carbon. These results may be due to the difference in roles and activation mechanisms for both of them as chemical reagents during the carbonization process. The activation by KOH under CO_2_ flow slightly improved the porous structure (specific surface area and pore volume), while for activation under N_2_, the pore volume and specific surface area were much improved. Figure 1 shows the KOH activation mechanism [55]. According to the KOH activation mechanism and earlier findings, we concluded that because KOH is a strong base, it can interact directly with carbon atoms and so catalyze dehydrogenation and oxidation processes, resulting in carbon consumption, tar suppression, and the formation of pores [56].

However, the random distribution of KOH inside the matrix of *Pisum sativum* pods may result in hyperactivation after heat treatment, collapsing the pore walls and resulting in the enlargement of micropores into mesopores. Additionally, the results in Table 1 show that ZnCl_2_ as an impregnation reagent under CO_2_ and N_2_ gases is more effective in pore development and in increasing specific surface area than KOH. Figure 2 illustrated the ZnCl_2_ activation mechanism. ZnCl_2_ is considered a Lewis acid, so it has a completely different activation mechanism from the KOH activation mechanism. ZnCl_2_ can act as a strong dehydrating agent, which can easily interact with the oxygen atom in OH functional groups in lignocelluloses substances (*Pisum sativum* pods), causing a dehydration reaction and dehydrogenation from the hydroaromatic structure. Furthermore, during the heat treatment, ZnCl_2_ fused and reacted with H_2_O, forming the zinc oxide chloride hydrate (Zn_2_OCl_2_·2H_2_O). Most importantly, due to increasing the temperature, the Zn_2_OCl_2_·2H_2_O decomposed and the ZnCl_2_ gas could evolve, which played an important role in the formation of many new sites, subsequent growth in the pore structure, and creation of the porosity in the final product. Additionally, ZnCl_2_ gas hinders the formation of tar inside the formed pores [57,58].

### 2.2. Characterization of Activated Carbon

#### 2.2.1. The X-ray Diffraction

Figure 1a shows the XRD patterns of raw pea pods impregnated with KOH (RPP-KOH) and activated carbon prepared by pyrolysis at 800 °C under flow of CO_2_ (AC-KOH). The XRD pattern for the sample RPP-KOH displays two small broad diffraction peaks around 24° and 44°, corresponding to the (002) and (100) diffractions for carbon. The XRD pattern for sample AC-KCO showed a broad diffraction peak located at 2*θ* = 24°, and another peak at 2*θ* = 44°, which became markedly sharper. According to this peak, it is possible that the activated carbons contain potassium compounds with high crystallinity after being activated with KOH. K metal was created mostly during the activation process at temperatures above 700 °C [59]. The XRD patterns of raw pea pods impregnated with ZnCl_2_ (RPP-ZnCl_2_) and its activated carbon prepared by pyrolysis at 800 °C under flow of CO_2_ (AC-ZnCl_2_-CO_2_) are shown in Figure 1b. The RPP-ZnCl_2_ sample showed a broad diffraction peak located at 2*θ* = 22° and a small intensity peak at 44°. The AC-ZnCl_2_-CO_2_ sample’s XRD pattern revealed a minor shift in the peak from 22° to 25°, indicating the presence of an amorphous structure with unevenly stacked carbon rings that are helpful for forming an adsorption gap. Additionally, the peak at 44° became somewhat sharp, indicating that AC-ZnCl_2_-CO_2_ had a more orderly structure than RPP-ZnCl_2_ and that the activation process may possibly promote the formation of graphite microcrystallites. Ultimately, the peaks between 10° and 20° are attributed to the presence of micropores and microcrystallinity in the synthesized activated carbon [60], which could be related to the graphite-like microcrystalline structure of multilayer stacks [55].

#### 2.2.2. FTIR- Spectra

Figure 2a–d and Figure 3a–c showed the FTIR spectra of raw pea pods (*Pisum sativum*) (RPP), the raw pea pods impregnated with a 2:1 ratio of ZnCl_2_ (RPP-ZnCl_2_) and with 10% KOH (RPP-KOH) samples, and activated carbon prepared by pyrolysis at 800 °C under flow of N_2_ and CO_2_ (AC-ZnN and AC-ZnCO), respectively. We observed that all the spectra in Figure 2a–d and Figure 3a–c are similar. The development of a broad band around 3290 cm^−1^ correlates to hydroxyl (OH)-stretching vibration functional groups, which vanished when the temperature was increased to 800 °C. At 2850 cm^−1^, the presence of a faint band suggests the presence of an aliphatic −CH-stretching vibration, which was totally abolished at 800 °C. The emergence of a very weak peak at about 2650 cm^−1^ following activation at 800 °C was demonstrated to correspond to the existence of stretching (C-H aldehydes). The appearance of a minor peak around 2300–2350 cm^−1^ that is linked to C≡C-stretching vibrations in alkyne groups (Figure 2c and Figure 3b). In the case of activated carbon formed by pyrolysis at 800 °C with a 2:1 ratio of ZnCl_2_ under CO_2_ flow, the steep peak at 1600 cm^−1^ correlates to C=C skeletal stretching of the aromatic rings, which was more pronounced than when the activated carbon was generated under N_2_ flow. Due to the decomposition of C−H bonds at the higher activation temperature (800 °C) to generate more stable aromatic C=C bonds, this peak may become sharper with ZnCl_2_ under CO_2_ flow. C=O, C–O of carboxylic groups, or in-plane vibration of O−H of carboxylic groups can all be blamed for the appearance of a peak at 1400 cm^−1^. During the aromatization of pea pods at 800 °C as the activation temperature, this peak was completely erased. The stretching vibration of the C–O group in carboxylic acids, alcohols, phenols, ethers, esters, or the P=O bond in phosphate esters matched to the broad peaks at around 1050–1200 cm^−1^. C−C stretching may be responsible for the peaks around 700–400 cm^−1^. The small absorption peak at 600 cm^−1^ could be attributed to Si-H-stretching vibrations or benzene polycyclic and C−H bending, which are also possible explanations. Following the activation procedure, the majority of the functional group’s adsorption peak was eliminated, and new peaks emerged. This could be due to the feedstock’s functional groups evaporating as volatile molecules when heated, indicating that the activation process has been completed properly [61].

#### 2.2.3. Scanning Electron Microscopy (SEM)

Figure 4a–f shows the SEM images of the microstructure of the *Pisum sativum* pods, which were impregnated with a 2:1 ratio of ZnCl_2_ and activated at different temperatures (600, 700, and 800 °C) under CO_2_ gas for 60 min. It can be seen from SEM images taken during the activation stage (700 °C) that several porous and hollow carbon holes were observed on the activated carbon^’^s surface. The external surface of the activated carbon has a porous structure with a large number of micropores because the most volatile organic materials evolved. Micrographs (~nm range) of activated carbon taken during experiments performed at carbonization temperatures 600 and 800 °C exhibit an extensive external surface with a very high number of micropores with diameters ranging from 69.95–48.88 nm, which act as channels for the microporous adsorbent. Obviously, activated carbon processes were successful, and the microporous structure was developed by chemical activation process with ZnCl_2_, so it had a large specific surface area, which was consistent with the BET value.

#### 2.2.4. The Thermogravimetric Analysis (TGA and DTA)

The TGA and DTA profiles of the raw pea pods (*Pisum sativum*) (RPP) and their impregnation with a 2:1 ratio of ZnCl_2_ (RPP-ZnCl_2_) and with 10% KOH (RPP-KOH) samples give an excellent vision about the range of carbonization temperatures needed to produce the activated carbon. The TGA and DTA curves of raw pea pods, as well as KOH- and ZnCl_2_-impregnated samples, are shown in Figure 5. Pyrolysis of raw pea pods involves four phases of deterioration. In the first phase, weight loss of around 4.15 percent is seen in the temperature range of 56.46 to 190 °C, which could be attributed to the release of water, such as moisture and bound water, as well as light volatile components. The second step shows a rapid decrease in the weight loss (approximately 44.07% at the temperature range from 192.98 to 275 °C), which could be attributed to depolymerization of hemicelluloses. The third step involves a gradual loss in weight at temperatures above 275 to 394.09 °C. However, there was a weight loss of 18.49%, which may be attributed to the decomposition of cellulose. The final step, which took place between 400 and 950 °C, showed a modest decreasing trend and a weight loss of 14.02 percent, which could be related to the degradation of the lignin moiety in pea pods as well as to the recombination of structure and synthesis of the fundamental carbon skeleton [62,63].

The DTA curve in Figure 5 shows a significant double maximum at 260 and 306.99 °C between 192.98 and 394.09 °C. This implies that the raw pea pods’ primary breakdown took place in this temperature range. The TGA/DTA curves during the pyrolysis process at a 10% weight ratio of KOH impregnation are shown in Figure 5. During the pyrolysis process, KOH-impregnated samples show different thermal properties than raw pea pods, which could be owing to the reaction between pea pods and K compounds. The first step of weight loss was identical to raw pea pods and occurred at 55.86–160 °C, which is slightly sooner than raw pea pods (56.46–190 °C). Dehydration and moisture evaporation may be to blame for the weight loss [64]. The second weight loss happened between 183.51 and 541.59 °C, resulting in a 49.01 percent weight reduction. This weight loss is mostly due to the breakdown of activating agencies at temperatures about 200 °C, in which KHCO_3_ rapidly degrades into H_2_O and CO_2_, resulting in increased weight loss [64] followed by hemicellulose and cellulose decomposition [65]. From 544–831 °C, the third weight loss of the KOH-impregnated sample occurred, with a small weight loss (12.84%), which can be attributed to the breakdown of K_2_CO_3_ and lignin [66]. The fourth weight loss of the KOH impregnated sample occurred at 831.70–950 °C, with a weight loss of 17.88%, which can be attributed to the ongoing degradation of lignin and K_2_CO_3_ as well as to the effect of burn-off at high temperatures, which causes the carbon skeleton to disintegrate.

It is worth noting that the mass continues to decrease from 700 to 950 °C due to the release of carbon dioxide and potassium sublimation [57]. This means that during the chemical activation of pea pods, the activation temperature should not exceed 800 °C. The largest rate of weight loss occurred at roughly 263.71, 699.96, and 852.36 °C, as seen in the DTG curve in Figure 5, and ended at around 986.95 °C. For ZnCl_2_-activated sample, the pyrolysis process involved four steps, as shown in Figure 5. In the first stage, at around 58.84–130 °C, the ZnCl_2_-activated sample experienced a weight loss of around 6.94%; this may be due to the released of H_2_O within the biomass. The second and third stages showed more gradual reductions in mass (42.14 and 33.16%, respectively) from 178.18 to 745.02 °C. Meanwhile, a wide range of the weight loss was due to degradation process of lignocellulose materials and the release of moisture from the solid-phase ZnCl_2_. The final stage showed a weight loss of 5.19% around 745.02–950 °C. The results were most likely of the total evaporation of the liquid phase from the ZnCl_2_ at a temperature >700 °C. At temperatures >800 °C, ZnO was reduced to metallic zinc [67]. The differential thermogravimetric analysis (DTA) of the ZnCl_2_-activated sample shown in Figure 5 showed that the maximum weight loss rate was observed at two basic peaks at 254.99 °C and 588.24 °C.

#### 2.2.5. Characterizations of Pore Structure

Different approaches (BET, t-Plot, MP, and BJH) were used to optimize the specific surface area of prepared activated carbons (Figure 6, Figure 7, Figure 8, Figure 9 and Figure 10). The N_2_ adsorption–desorption isotherms of activated carbon made from pea pods impregnated with 10% KOH and 2:1 ZnCl_2_ for 1 h at various activation temperatures of 600, 700, and 800 °C under N_2_ and CO_2_ gases are shown in Figure 6a–d. In the comparison of the adsorption isotherms of different activated carbons, it is revealed in Figure 6 that the adsorption isotherms of the different activated carbons are typical type I microporous carbons according to IUPAC classification [22,23,68]. The nitrogen adsorption isotherm height of activated carbon (AC-KOH) at 800 °C is far more than that of activated carbon (AC-KOH) at 600 and 700 °C under N_2_ gas (Figure 6a), while under CO_2_ gas, the nitrogen adsorption isotherm height of activated carbon (AC-KOH) at 600 °C is far more than that of activated carbon (AC-KOH) at 700 and 800 °C (Figure 6b). Figure 6a, showing the AC-KOH samples at different temperatures from 600–800 °C under N_2_, showed a small adsorption quantity compared to the AC-ZnCl_2_ samples under the same conditions (Figure 6c). As illustrated in Figure 6b, the AC-KOH samples at 600, 700, and 800 °C in the presence of CO_2_ exhibit a negligible capacity for N_2_ adsorption compared to approximately one fourth of the AC-ZnCl_2_ samples in the same conditions (Figure 6d). This finding reveals that when the impregnation reagent is changed from KOH to ZnCl_2_, the porosity of activated carbon increases dramatically. As previously stated, Figure 6c,d demonstrates an increase in the maximum adsorption quantity when activated carbon made from pea pods soaked with 10% KOH is used (Figure 6a,b). However, the isotherms’ knees are quite abrupt at low relative pressures (less than 0.05), and the adsorption capacity of samples (AC-ZnCl_2_) for N_2_ is close to saturation at 0.1. When the relative pressure is more than 0.1, the adsorption quantity increases very slowly. This implies that all samples activated with AC-ZnCl_2_ possessed well-developed micropores. It is worth noting that a greater number of pores are formed in superficially modified activated carbon (AC-ZnCl_2_) at 800 °C activation temperature under CO_2_ gas than in other activated carbons (AC-ZnCl_2_) at the same activation temperature under N_2_ gas (AC-KOH) or at different activation temperatures from 600–800 °C under N_2_ and CO_2_ gases. We may conclude from the aforementioned data that the most effective parameter, among others, was the impregnation reagent (ZnCl_2_) and the type of medium gas (CO_2_).

Table 1 lists the pore architectures of activated carbon generated from pea pod agricultural waste material using KOH and ZnCl_2_ activation at 600–800 °C in the presence of N_2_ and CO_2_ gases. For impregnation of potassium-hydroxide-activated carbon (AC-KOH) under CO_2_, the BET specific surface area was found to be 357.82, 87.70, and 241.84 m^2^/g and the mean pore diameter was 2.068, 2.974, and 2.462 nm while the total pore volume was 0.185, 0.065 and 0.149 cm^3^/g at 600, 700, and 800 °C, respectively. On the other hand, for impregnation of potassium-hydroxide-activated carbon under N_2_, the BET specific surface area (Appendix A) was found to be 197.33, 716.28, and 914.58 m^2^/g and the mean pore diameter was 2.33, 1.94, and 1.77 nm while the total pore volume was 0.115, 0.347 and 0.405 cm^3^/g at 600, 700, and 800 °C, respectively. Interestingly, when compared to CO_2_, the AC-KOH exhibited greater improvements in specific surface area, mean pore diameter, total pore volume, and specific surface area per unit volume. These novel microporosity creations caused by the volatile evolution process, carbon oxidation process, and carbon gasification process may be responsible for this (AC-KOH, under N_2_). During the chemical activation procedure for the samples of (AC-KOH) under CO_2_, a widening of microporosity into the region of mesoporosity was observed. This could be due to the release of tar caused by the extreme activation of activated carbon, which resulted in an expansion of porosity in the activated carbon layer. When the temperature of activated carbon impregnated with ZnCl_2_ (AC-ZnCl_2_) was raised from 600 °C to 800 °C, the specific surface area of the activated carbon increased from 1039 m^2^/g to 1084 m^2^/g. Additional results revealed that when the temperature was raised from 600 to 800 °C, the total pore volume decreased somewhat, from 0.498 to 0.479 cm^3^/g, and the microporosity was determined to be 1.915, 1805, and 1.767 nm in diameter. Clearly, the pore architectures of activated carbon impregnated with ZnCl_2_ were significantly improved when exposed to CO_2_.

As the temperature was raised from 600 to 800 degrees Celsius, the specific surface area increased dramatically, rising from 1228.00 to 1299.40 m^2^/g as a result of the formation of a large number of new micropores. Additionally, when the temperature was raised from 600 to 800 °C, the total pore volume increased marginally, from 0.552 to 0.618 cm^3^/g, despite the fact that the microporosity was 1.799, 1.859, and 1.901 nm. Finally, we conclude from these findings that activation with ZnCl_2_ under both gas conditions resulted in a considerable and obvious improvement in pore architecture (Appendix A). Additionally, the t-plot approach (Appendix A) was used to determine the nitrogen adsorption isotherm, in which the adsorption amount is given as the thickness of the adsorption layer (*t*). In general, t-plots can be classified into three types. However, this classification is arbitrary, as there is no type established by the IUPAC as there is with adsorption isotherms. However, as seen in Figure 8, the t-plot has two distinct slopes, one of which is a steep slope going through the initial point and the other of which is a more gradual slope, indicating that the adsorbent has micropores of uniform size. The amount of adsorption grows dramatically during the early stages of adsorption due to adsorption into micropores, while the thickness of adsorption does not increase as much. Once adsorption into micropores is complete, no further adsorption occurs on the surface.

As shown in Table 1, the pore surface area and volume were somewhat different than those determined by BET; this result suggested that the maximum specific surface area of micropores is expected to be between 1300 and 1500 m^2^/g. It is worth noting that the average pore diameter (2t) was significantly different than that determined by BET. Generally, the value (2t) produces incorrect analysis findings if the pore size is more than two layers. Additionally, when the 2t value is less than 0.7 nm, as it is with micropore filing, one can obtain an approximate indication of the size of the pores, but there is no numerical value to rely on. Because the MP-plot (Appendix A) is a derivation of the t-plot, the MP-plot’s results closely match those of the t-plot.

The mesopore and micropore size distributions of activated carbons generated by KOH at various activation temperatures (600, 700, and 800 °C) in the presence of N_2_ and CO_2_ are depicted in Figure 9a,b, respectively. Under N_2_ rather than CO_2_, activated carbons (AC-KOH) are dominated by micropores, and all activated carbons have a distribution peak about 0.6 nm.

As illustrated in Appendix A, the samples (AC-KOH) at 700 and 800 °C have a similar number of micropores, whereas the samples (AC-KOH) at 600 °C have fewer micropores. This demonstrates that changing the activation temperature has a significant effect on micropore formation. In comparison, Appendix A demonstrates that the sample (AC-KOH) heated to 600 °C under CO_2_ has a greater number of micropores than the samples (AC-KOH) heated to 700 and 800 °C. This suggests that switching to CO_2_ as the gas medium has a detrimental influence on the formation of micropores. The micropore size distributions of activated carbons generated with ZnCl_2_ at various activation temperatures (600, 700, and 800 °C) in the presence of N_2_ and CO_2_ are shown in Figure 9c,d. As shown in Appendix A, all activated carbon samples (AC-ZnCl_2_) have micropore sizes ranging from 0.4 to 1.1 nm and a distribution peak at 0.6 nm. We must emphasize that all samples (AC-ZnCl_2_) produced in N_2_ exhibit nearly identical micropores. As shown in Appendix A, a similar trend is observed for (AC-ZnCl_2_) under CO_2_, with the sample (AC-ZnCl_2_) at 600 °C having more developed micropores than the samples (AC-ZnC_2_) at 700 and 800 °C, indicating that the micropores developed more with a decrease in the activation temperature and using CO_2_ gas as an activation reagent. According to Figure 7a, all samples of (AC-KOH) under N_2_ had micropores with a radius of 2 to 3 nm and a distribution peak near 2.5 nm. At 600, 700, and 800 °C, the integrated pore volume (*V*p) was 0.0189, 0.0724, and 0.0376 cm^3^ g^−1^, respectively (Table 1). The pore size distribution of (AC-KOH) produced under CO_2_ is between 2 and 4.5 nm, with distribution peaks around 2.5 nm (Figure 7b). At 600, 700, and 800 °C, the integrated pore volume (*V*p) was 0.0280, 0.0190, and 0.0199 cm^3^ g^−1^, respectively (Table 1). According to Figure 7c, all samples of (AC-ZnCl_2_) in N_2_ had micropores with a radius of 1.5 to 2.5 nm and a distribution peak around 2 nm. At 600, 700, and 800 °C, the integrated pore volume (*V*p) was 0.0690, 0.0495, and 0.0433 cm^3^ g^−1^, respectively (Table 1). All (AC-ZnCl_2_) produced under CO_2_ had pore sizes ranging from 1 to 2.5 nm, with distribution peaks around 2 nm (Figure 7d). At 600, 700, and 800 °C, the integrated pore volume (Vp) was 0.0529, 0.0516, and 0.0581 cm^3^ g^−1^, respectively (Table 1). It appears from this result that the pore size of (AC-ZnCl_2_) generated under CO_2_ and N_2_ conditions can give a large specific surface area, making it well-suited for the adsorption and removal of pollutants from water and wastewater.

### 2.3. Acid Orange 7 Dye Adsorption Study

The selective activated carbon soaked with 2:1 ZnCl_2_ and heated to 800 °C in the presence of CO_2_ gas was utilized as an adsorbent to remove AO7 dye from water based on a general test comparing all the manufactured activated carbons under various conditions. The effects of pH, contact time, initial dye solution concentration, and adsorbent dose at room temperature were explored in order to design viable wastewater adsorption systems. Adsorption isotherms and kinetic models were also investigated in this study.

#### 2.3.1. Effect of Solution pH on the Adsorption Process

In the absorption process, the pH of a solution is believed to be a significant parameter for predicting the adsorption of dyes onto adsorbents during the adsorption procedure. In part, this is because the acidity or alkalinity (pH) of the solution has an effect on its chemistry and, subsequently, on the interaction between its functional group and the dye. The effect of solution pH on AO7 dye removal was investigated using a 100 mg/L initial dye concentration and pea pod activated carbon (AC-ZnCl_2_) prepared at 800 °C under CO_2_ gas (1.0 g/L) as an adsorbent, with the pH of the solution being varied from 1 to 11 at room temperature (Figure 8). Figure 8 shows that AO7 dye removal efficiency decreased as the pH increased; however, when the pH of the solution reached 5.12, the removal of dye started to increase slightly until pH 7 and then started to decrease again. The maximum percentage uptake was obtained at pH 1.51 (97.4%), while the uptake percentage of AO7 dye at pH 5.12 and 9.97 had the minimum values (62.0 and 66.6%, respectively). It appears that the changing of pH value led to the formation of a different carbon surface charge. In particular, at lower pH values, protonation onto the surface of the adsorbent occurred. It is worth noting that the activated carbon (AC-ZnCl_2_) was indeed essentially microporous and had a higher surface area, implying that their overall porosity was available for dye adsorption because the activated carbon (AC-ZnCl_2_) allowed H^+^ ions to enter the micropores structure, forming strong electrostatic attraction between negatively charged AO7 dye anions and positively charged adsorption sites and resulting in an increase in adsorption efficiency.

#### 2.3.2. Impact of Contact Time and Initial Dye Concentration on the Adsorption Process

Finally, the adsorption capacity is highly dependent on the time of adsorption. Thus, the time course of the percentage removal of AO7 dye adsorbed on the AC-ZnCl_2_ (2.5 g/L) was investigated using AO7 dye at various starting concentrations (100–400 mg/L) at ambient temperature (Figure 9a). At an early stage (15 min), the percentage of AO7 dye removed increased rapidly and the initial adsorption rate onto the activated carbon surface via the interconnected micropores was extremely high—on the order of 94.76, 93.75, 91.75, 81.61, and 79.24 percent for initial concentrations of 100, 150, 200, 300, and 400 mg/L, respectively (Figure 9a). Additionally, notice that prior to reaching equilibrium (15–60 min), there is a small increase in AO7 dye removal efficiency with increasing contact time. After approximately 60 min, equilibrium adsorption was achieved for all initial dye concentrations (100–400 mg/L), with removal percentages of 99.11, 98.46, 98.24, 96.77, and 96.53%, respectively, and the maximum percentage removal of dye (100 percent) was observed after 180 min for all initial concentrations (Figure 9a). The initial concentration acts as a powerful motivator for overcoming mass transfer resistances between the solid surface and liquid phase. The relationship between starting dye concentrations of 100–400 mg/L, different adsorbent dosages of AC-ZnCl_2_ (0.75–2.5 g/L), and adsorbent quantity (*Q_e_*) was examined. The investigation’s findings are depicted in Figure 9b.

The quantity of adsorbent of AO7 dye at equilibrium (*Q_e_*) increases from 133.23 to 467.16 mg/g with increasing starting concentrations from 100 to 400 mg/L in the current investigation at adsorbent dosage 0.75 g/L. With an increase in the starting concentration from 100 to 400 mg/L, the quantity of adsorbent of AO7 dye at equilibrium (*Q_e_*) increased from 99.97 to 356.47, 66.65 to 248.58, 49.94 to 198.13, and 39.99 to 159.15 mg/g at adsorbent dosages of 1.0, 1.5, 2.0, and 2.5 g/L of AC-ZnCl_2_, respectively. With a decrease in the adsorbent doses from 0.75 to 2.5 g/L, the *Q_e_* increased dramatically from 159.15 to 467.16 mg/g at an initial concentration of 400 mg/L. It is widely assumed that the mechanism for dye adsorption is linked to the pore structures of activated carbon, specifically activated carbon prepared by ZnCl_2_ and CO_2_ activation, which showed the ability to exhibit a large number of micropores and a large quantity of oxygen-containing functionalities of C−O groups in carboxylic acids, alcohols, phenols, ethers, esters, or the P=O bond in phosphate esters, which can provide active sites for dyes (Figure 2 and Figure 3).

#### 2.3.3. Adsorption Isotherms

To describe adsorption mechanisms, adsorbate interaction with adsorbent, and adsorbate distribution between the liquid and solid phases at equilibrium, five isotherm equations were fitted to the experimental data. The adsorption of AO7 dye onto AC-ZnCl_2_ was examined using the Langmuir, Freundlich, Harkins–Jura, Halsey, generalized, and Tempkin isotherm models [69]. The slopes, intercepts, and correlation coefficients (*R*^2^) were used to derive the isotherm model parameters, which are listed in Table 2. Based on the Langmuir constant, *K_L_* and *R*^2^ were both larger at the higher adsorbent dose (2.5 g/L), indicating a stronger interaction between the adsorbent and adsorbate. The Langmuir isotherm equation was found to be the best fit for the adsorption equilibrium data of the activated carbon (AC-ZnCl_2_) sample, with the highest *R*^2^ value being ~1.000, implying that this isotherm provides the most accurate representation of the experiment data and indicating the homogeneous nature of activated carbon. The findings also showed that monolayer physical adsorption could effectively explain the adsorption of AO7 dye onto activated carbon from aqueous solutions and that adsorbate molecules on the surface of the AC-ZnCl_2_ are energetically equal. Furthermore, the quantity of adsorption equivalent to complete monolayer coverage calculated using the Langmuir model (*Q_m_*) was 473.93 mg/g, which is extremely close to the 467.16 mg/g observed from practical measurements at 0.75 g/L (Table 2).

These results can be explained as follows: the carbon activated by ZnCl_2_ under CO_2_ at 800 °C, which has the largest surface area (1300 to 1500 m^2^/g) and highest total pore volume (0.6176 cm^3^/g) (Table 1) and also showed the highest adsorption capacity (467.16 mg/g) of AO7 dye, had an excellent fit with the Langmuir isotherm due to the predominant presence of micropores with small diameters, which provided more channels and binding sites for AO7 dye onto the surface of the adsorbent and more difficulty for the penetration of AO7 dye ions as a bulk component, indicating the occurrence of physical adsorption. In addition, there were more aromatic C=C groups in the activated carbon (AC-ZnCl_2_) prepared under CO_2_ activation than that prepared under N_2_, and the π–π dispersion interaction between the aromatic structures of AO7 dye and AC-ZnCl_2_ prepared under CO_2_ activation can also increase the adsorption performance (Figure 2d).

#### 2.3.4. Adsorption Kinetics

The rate kinetics of the AO7 dye adsorption on the activated carbon AC-ZnCl_2_ in relation to residence time was examined to regulate and understand the adsorption dynamic mechanisms and solute absorption at the solid–solution interface, including the diffusion process. The models used were pseudo-first-order, pseudo-second-order, Elovich, intraparticle, and film diffusion. The correlation coefficient was used to express the similarity of experimental and model-predicted data (*R*^2^).

##### The Pseudo-First-Order and Pseudo-Second-Order Kinetic Models

The kinetic data for AO7 dye adsorption at various initial dye concentrations (100–400 mg/L) and varied adsorbent dosages of AC-ZnCl_2_ (0.75–2.5 g/L) were calculated from the associated plots and are summarized in Table 3. Figure 11a,b illustrates the graphs of The pseudo-first-order and pseudo-second-order kinetic models, respectively. According to the results, the correlation coefficients for the pseudo-second-order kinetic model (1.000) were significantly higher than those for the pseudo-first-order kinetic model; secondly, increasing the initial dye concentration from 100 to 400 mg/L at 0.75 g/L results in an increase in the calculated *Q_e_* values from 138.89 to 500.00 mg/g. Finally, because the experimental adsorption capacity (467.16 mg/g) was nearly identical to that predicted by the pseudo-second-order kinetic model (500.00 g/L), it was concluded that the studied adsorption system obeys the pseudo-second-order kinetic model and that physisorption may be the rate-limiting step [69].

##### Elovich Kinetic Model

This model is frequently used to describe the chemisorption mechanism, which may be responsible for controlling the rate of adsorption. Figure 11c illustrates the Elovich kinetic data obtained in this investigation for the adsorption of AO7 dye starting concentrations (100–400 mg/L) at adsorbent doses (2.5 g/L) from aqueous solution at room temperature onto AC-ZnCl_2_. The linear connection depicted in Figure 11c has a slope of 1/*β* and an intercept of 1/*β* ln(*αβ*) (Equation (15)). The desorption constant (*β*), the initial adsorption rate constant (*α*), and the correlation coefficient (*R*^2^) are shown in Table 4 and are all derived from the Elovich rate Equation (15). Despite the fact that the desorption constant (*β*) dropped when the initial concentrations of AO7 dye increased from 100 to 400 mg/L and the initial adsorption rate (*α*) increased, the correlation coefficient (*R*^2^) is lower than that predicted by the intraparticle and film diffusion kinetic models. This result shows that the Elovich model was not the best model.

##### Intraparticle and Film Diffusion Kinetic Models

Additionally, to investigate the likelihood of dye ion molecules being transferred from the bulk solution into the pores or onto the adsorbent’s outer surface via film diffusion or intraparticle diffusion, the rate-limiting stage in the adsorption process was used. In essence, the slower of the two steps will serve as the rate-limiting step. The intraparticle and film diffusion rate constants (*K*_dif_ and *K_FD_*, respectively) and correlation coefficient (*R*^2^) calculated from the intraparticle and film diffusion rate equations are listed in Table 4. Figure 11d,e illustrates the intraparticle and film diffusions of AO7 dye at starting concentrations (100–400 mg/L) and adsorbent doses (2.5 g/L) onto AC-ZnCl_2_. As illustrated in Figure 11d,e, the plots in both models are linear and do not pass through the origin.

As shown in Table 3, the correlation coefficient (*R*^2^) was 1.000 for all initial concentrations of AO7 dye (100–400 mg/L) at various adsorbent dosages (0.75–2.5 g/L) of AC-ZnCl_2_) in the film diffusion model. However, the correlation coefficient (*R*^2^) was lower in the intraparticle diffusion model. According to the results, the film diffusion technique was used to modulate the adsorption rate of AO7 dye ions onto the AC-ZnCl_2_. The linear deviation from the origin or near saturation during the adsorption process could be due to the presence of an initial boundary layer of H^+^ ions that penetrated the micropore and act as a barrier to the intraparticle diffusion mechanism, or it could be due to the difference in mass transfer rates between the initial and final stages of adsorption [70,71].

**Table 3 molecules-27-04840-t003:** The findings of pseudo-first-order and pseudo-second-order adsorption of AO7 dye at various initial concentrations (100–400 mg/L) at 25 ± 2 °C onto pea pod activated carbon generated at a carbonization temperature of 800 degrees Celsius in the presence of CO_2_ using ZnCl_2_ as the activating agent.

Parameter			First-Order Kinetic Model			Second-Order Kinetic Model			
AC (g/L)	AO7 Dye (mg L^−1^)	*q_e_* (exp.)	*q_e_* (calc.)	*k*_1_ × 10^3^	*R* ^2^	*q_e_* (calc.)	*k*_2_ × 10^3^	*h*	*R* ^2^
0.75	100	133.23	101.81	46.52	0.989	138.89	0.99	19	1.000
	150	198.77	144.18	38.69	0.992	212.77	0.48	22	0.999
	200	264.77	185.65	28.79	0.898	285.71	2.68	219	0.999
	300	379.51	195.48	24.87	0.988	400.00	0.22	36	1.000
	400	467.16	223.87	19.81	0.963	500.00	0.15	38	0.999
1.0	100	99.97	36.53	39.61	0.981	103.09	2.26	24	1.000
	150	149.71	64.46	37.77	0.968	156.25	1.05	26	1.000
	200	199.26	81.49	22.80	0.853	208.33	0.50	22	1.000
	300	287.24	122.01	21.88	0.994	303.03	0.32	29	1.000
	400	356.47	138.90	16.81	0.967	370.37	0.24	33	0.999
1.5	100	66.65	22.09	47.44	0.969	68.03	4.89	23	1.000
	150	99.89	47.40	46.29	0.995	103.09	2.09	22	1.000
	200	133.05	47.96	34.08	0.971	136.99	1.37	26	1.000
	300	195.44	87.24	25.56	0.912	204.08	0.58	24	0.999
	400	248.58	96.32	21.88	0.974	263.16	3.80	263	1.000
2.0	100	49.99	10.20	43.07	0.965	50.76	9.88	25	1.000
	150	74.96	17.96	42.38	0.974	76.34	4.82	28	1.000
	200	99.89	27.52	37.08	0.927	102.04	2.61	27	1.000
	300	148.75	50.69	31.78	0.965	153.85	1.24	29	1.000
	400	198.13	84.94	27.18	0.979	208.33	0.61	26	1.000
2.5	100	39.99	3.79	38.69	0.981	40.16	26.72	43	1.000
	150	59.99	7.92	38.92	0.995	60.61	12.43	46	1.000
	200	79.97	11.87	35.93	0.986	80.65	7.39	48	1.000
	300	119.57	26.85	31.78	0.968	121.95	2.62	39	1.000
	400	159.15	40.96	31.09	0.970	163.93	1.50	40	1.000

**Table 4 molecules-27-04840-t004:** The results of Elovich, intraparticle diffusion, and film diffusion kinetic models used to predict the adsorption of AO7 dye (100–400 mg/L) onto pea pod activated carbon produced at 800 °C carbonization temperature under CO_2_ using ZnCl_2_ as the activating agent.

AC (g/L)	AO7 conc.	Elovich			Interaparticle Diffusion			Film Diffusion		
		*β*	*α*	*R* ^2^	*K_dif_*	*C*	*R* ^2^	*K_FD_*	*C*	*R* ^2^
0.75	100	0.0542	2.23 × 10^2^	0.927	0.15	1.82	0.887	0.05	0.44	1.000
	150	0.0322	1.55 × 10^2^	0.960	0.17	1.93	0.927	0.04	0.47	0.999
	200	0.0245	1.75 × 10^2^	0.993	0.18	2.03	0.982	0.33	0.41	0.999
	300	0.0179	3.44 × 10^2^	0.990	0.17	2.21	0.975	0.26	0.63	0.999
	400	0.0143	1.42 × 10^1^	0.977	0.18	2.27	0.971	0.21	0.68	1.000
1.0	100	0.1067	3.60 × 10^3^	0.906	0.09	1.80	0.871	0.04	1.03	0.999
	150	0.0525	4.46 × 10^2^	0.884	0.13	1.90	0.846	0.04	0.67	1.000
	200	0.0356	2.51 × 10^2^	0.974	0.16	1.96	0.951	0.03	0.67	0.997
	300	0.0250	3.61 × 10^2^	0.982	0.16	2.11	0.966	0.02	0.84	0.997
	400	0.0213	5.22 × 10^2^	0.968	0.16	2.20	0.969	0.02	0.82	0.999
1.5	100	0.1972	2.43 × 10^4^	0.869	0.07	1.67	0.831	0.05	1.07	0.996
	150	0.0966	1.47 × 10^3^	0.889	0.10	1.78	0.851	0.05	0.80	0.999
	200	0.0668	9.62 × 10^2^	0.891	0.12	1.88	0.855	0.04	0.82	0.998
	300	0.0456	1.04 × 10^3^	0.988	0.13	2.02	0.978	0.03	0.76	0.995
	400	0.0295	3.76 × 10^2^	0.974	0.15	2.07	0.950	0.03	0.55	0.999
2.0	100	0.4188	2.66 × 10^7^	0.892	0.05	1.60	0.864	0.05	1.50	0.999
	150	0.2040	1.97 × 10^5^	0.873	0.06	1.74	0.843	0.04	1.40	0.998
	200	0.1111	5.47 × 10^3^	0.874	0.09	1.81	0.837	0.04	0.85	0.998
	300	0.0643	1.87 × 10^3^	0.901	0.11	1.94	0.872	0.03	0.90	0.997
	400	0.0434	8.93 × 10^2^	0.975	0.13	2.02	0.960	0.03	0.83	0.998
2.5	100	1.1502	7.05 × 10^17^	0.921	0.02	1.56	0.897	0.04	2.37	0.999
	150	0.6039	7.34 × 10^13^	0.969	0.03	1.72	0.947	0.04	2.08	0.999
	200	0.3456	2.43 × 10^10^	0.955	0.03	1.83	0.932	0.04	1.85	0.999
	300	0.1163	8.26 × 10^4^	0.863	0.07	1.93	0.828	0.03	1.50	0.999
	400	0.0714	1.05 × 10^4^	0.885	0.09	2.02	0.856	0.03	1.07	1.000

#### 2.3.5. Adsorption Mechanism

As previously stated, the adsorption mechanism of the AO7 dye onto the AC-ZnCl_2_ surface can be extrapolated from pore structure studies, FTIR, adsorption isotherms, and kinetics model results. To begin, the presence of a large number of new micropores enhanced the diffusion of the H^+^ ions, forming the film via electrostatic attraction caused by functional groups such as C−H (aldehydes), C−O group in carboxylic acids, alcohols, phenols, ethers, and esters, or the P=O bond in phosphate esters. Secondly, the π–π interactions that occur between the π–electron system (C=C skeletal stretching of the aromatic rings) of the AC-ZnCl_2_ structure, and the aromatic rings of the AO7 dye molecules did the same (Figure 12).

#### 2.3.6. Comparison of the *Q_m_* of AO7 Dye onto Different ACs

Table 5 compares the maximum adsorption capacity (*Q_m_*) of several types of adsorbents employed to remove AO7 dye from its aqueous solution to the present work. When compared to other adsorbents reported in the literature, the *Pisum sativum* pea pod AC-ZnCl_2_ demonstrated a greater affinity for AO7 dye removal, with a *Q_m_* of 473.93 mg/g (Table 5). The increased adsorption capacity of AC-ZnCl_2_ may be a result of the micropores produced on its surface. As a result, AC-ZnCl_2_ derived from pea pod solid waste can be employed as a promising adsorbent for extracting AO7 dye from wastewater.

#### 2.3.7. Desorption Studies

The desorption study of desorbed AO7 dye on AC-ZnCl_2_ was carried out by using 1% NaOH (*w*/*v*). It was found that the AC-ZnCl_2_ can be used as an adsorbent for more than six cycles without much change in adsorption efficiency. The amount of desorbed AO7 dye was determined by using the following Equation (1):(1)$$Q_{d}=\frac{\left(C_{d}-C_{s}\right)V}{W}$$
where 𝐶_𝑑_ is concentration of AO7 dye after desorption and 𝐶_𝑠_ is concentration of dye after adsorption. The results of adsorption and desorption cycles are represented in Figure 13.

### 2.4. Optimization Study

The design matrix was used to investigate the interaction effects of three important factors, including contact time, activated carbon dose, and initial AO7 dye concentration on the removal of AO7 dye. The experimental design and the responses are shown in Table 6. Based on the obtained results, the following polynomial Equations (2) and (3) for AO7 dye removal were developed:Removal % for Coded Factors = 94.79 + 10.51A + 7.25B − 5.69C − 2.99AB + 1.56AC + 2.7BC − 7.13A^2^ − 3.3B^2^ − 0.1193C^2^(2)
Removal % for Actual Factors = 58.32749 + 0.605691 Time + 21.57175 Dose − 0.082057 Conc. − 0.065112 Time × Dose + 0.000198 Time × Conc. + 0.020565 Dose × Conc. − 0.002585Time^2^ − 4.31590 Dose^2^ − 0.000053 Conc.^2^(3)

The equation expressed in terms of actual factors enables prediction of the reaction at specified levels of each factor. For each factor, the levels should be indicated in their original units. This equation should not be utilized to calculate the relative impact of each element because the coefficients have been scaled to account for the units of each factor and the intercept is not in the design space’s center. 

A correlation between predicted and actual adsorption (%) of AO7 dye on activated carbon is shown in Figure 14. It is clear from the figure that there is good agreement between the experimental values and the predicted model, which is validated by the high value of the correlation coefficient (*R*^2^ = 0.9785). The ANOVA is given in Table 7 and is used to predict the cubic, individual, and interaction effects of the independent variables on the adsorption of AO7 dye on activated carbon. The results suggest that the quadratic model (*p*-value < 0.05) has a significant contribution. The determination coefficient described the standard of the polynomial model as a basis of the extent of deviation through the mean elucidated by the model, and the values of Adj-*R*^2^ = and *R*^2^ = show a good correlation between the predicted and exponential data [78,79,80,81]. The Predicted *R*² of 0.7850 is in reasonable agreement with the Adjusted *R*² of 0.9591; i.e., the difference is less than 0.2. Adeq Precision measures the signal to noise (S/N) ratio. A ratio greater than 4 is desirable. The S/N value of 26.139 indicates an adequate signal, which shows a significant RSM-model signal that can be utilized for navigating the design [78,82].

#### Simultaneous Effects of Interactive Adsorption Variables

Three-dimensional surface plots present the effects and interactions of independent variables, namely contact time, activated carbon dose, and initial dye concentration, on the removal percentages of AO7 dye (Figure 15) as the responses.

The interaction of initial ion concentration and adsorbent dosage shown in Figure 15c indicates the significant influence of both factors on the removal of AO7 dye. The removal percentages increased with increasing adsorbent dosage. This result was due to the presence of additional active sites and a large adsorbent surface area that is readily available for adsorption [83,84]. The removal percentage was reduced by increasing the initial ion concentration, as shown in Figure 15b. This finding might be due to the limited active sites on the adsorbent surface at high AO7 dye concentrations [85]. The removal percentage was increased by increasing the residence time from 40 min to 60 min. These results confirmed that the initial adsorption rate was very rapid due to the availability of a large surface area and the presence of unused sites on the activated carbon surface [86]. The slowing down of dye removal might be due to the difficulty of reaching the re-maining vacant sites. To optimize and validate the predicted mathematical model, a complementary statistical design calculation was performed under the same experi-mental conditions; Figure 16 shows that the higher desirability value obtained from the results of the mathematical model is equal to 0.982. Using these conditions, the maximum removal (%) (Removal (%) = 99.895%, experimental) obtained corresponds to the contact time: 95.7 min, AC dose 2 g/L and initial AO7 dye concentration of 100 mg/L.

## 3. Materials and Methods

### 3.1. Chemicals and Reagents

The raw material (*Pisum sativum* pea pods) employed in the present investigation as the raw material for the synthesis of activated carbon was obtained from a local market. After being scrubbed with tap water and distilled water to remove dirt and dust, these pea pods were sun-dried for two weeks before being used. Subsequently, the pea pods were powdered using a rotary mill at high speed and sieved through size <200 meshes. The powdered pea pods were kept in a cool, dry place until they were needed again. Potassium hydroxide (KOH) and zinc chloride (ZnCl_2_) were purchased from El-Nasr Company, Egypt. The hydrochloric acid (HCl) used in this experiment was obtained from Sigma-Aldrich in the United States. It was unnecessary to purify the Acid Orange 7 (AO 7) dye (C.I. 15510; chemical formula, C_16_H_11_N_2_NaO_4_S; molecular weight 350.33 g mol^−1^), which was obtained from Sigma-Aldrich in the United States. A stock solution of AO7 dye (1000 mg L^−1^) was prepared by dissolving the required amount of AO7 dye in distilled water and allowing the solution to stand for 24 h. The stock solution was diluted with distilled water to the desired concentrations to make all of the relevant working solution concentrations. All chemicals utilized were of analytic grade and had not undergone any further purification before being used in this study.

### 3.2. Preparation of Activated Carbon

#### 3.2.1. Impregnation with ZnCl_2_

Sieved pea pod powder was soaked in distilled water, impregnated with ZnCl_2_ in a 2:1 ratio (pea pods to ZnCl_2_, weight-to-weight), then baked until dry at 105 °C. Successively dried impregnated mixture was placed in an alumina boat, loaded in a horizontal tube furnace, heated at a ramp rate of 10 °C min^−1^, and kept at 600, 700, and 800 °C each for 60 min under a N_2_ atmosphere; the entire pyrolysis process was repeated under a CO_2_ atmosphere, for which the gas flow rate was 5 L min^−1^. Then, the furnace temperature was cooled to room temperature, and the carbonized sample was removed from the furnace and boiled in 2 N HCl for 2 h to remove residual organic and mineral matters. The filtrate was then rinsed many times with warm distilled water to bring the pH back to a neutral range before being dried in an oven at 105 °C for 24 h to remove any remaining contaminants. Activated carbon samples were milled and sieved using a mesh size <60 μm to obtain homogenous particle size.

#### 3.2.2. Impregnation with KOH

The sieved pea pod powder was impregnated with 10% KOH in distilled water. The impregnated mixture was dried at 105 °C in an oven for 24 h. Dried impregnated mixture was subjected to carbonization in an alumina boat in a horizontal tube furnace heated at a ramp rate of 10 °C min^−1^ and kept at 600, 700, and 800 °C each for 60 min under a N_2_ atmosphere; the pyrolysis process was repeated under a CO_2_ atmosphere. The carbon samples were cooled in a nitrogen or carbon dioxide environment, then removed from the furnace and repeatedly washed with distilled water until the filtrate’s pH was neutral. A 2 h reflux in 2N HCl was used to remove organic and mineral debris from the carbon, followed by filtration and washing with deionized water to bring it to a pH neutral value of around 7. 24 h. Oven drying at 105 °C produced the desired activated carbon. Activated carbon samples were milled and sieved using a mesh size <60 μm to obtain homogenous particle size.

### 3.3. Characterization of the Synthesized Activated Carbons

Brunauer–Emmett–Teller (BET), t-Plot, MP, and BJH methods were used for the determination of specific surface area (*S*_BET_), total pore volume, and mean pore size. They were estimated by the volume of nitrogen gas adsorbed to the activated carbon surface per unit mass of the measured samples using BELSORP Mini-II equipment, BEL Japan with a N_2_ adsorption at 77 K as adsorption temperature and saturated vapor pressure of 89.62 kPa. The samples were pretreated at 300 °C under flow of nitrogen gas. The micropore volume and total pore volume were determined using the *t*-plot method. The volume of the mesopores (V_MesoP_) was calculated using the difference between total pore volume (V_TP_) and total micropore volume (V_mP_). The average pore diameter (APD) was calculated using the 4V_TP_/S_BET_ ratio. The functional groups on the activated carbon surface were examined by Fourier transform infrared spectroscopy (FTIR) analysis that was performed on the activated carbon using a Bruker Model Vertex 70 FTIR spectrometer coupled to a Platinum ATR unit using a spectral range of 4000–400 cm^−1^, Bruker, Germany. Thermal analyses were performed using a SDT 650 TA Instrument—Waters LLC. The range was from 50 to 900 °C under a N_2_ flow of 100 mL min^−1^. The porosity and the surface morphology of the activated carbons obtained in this study were examined by SEM analysis performed using an FEI Quanta 250 FEG Scanning Electron Microscope using 500 KV HV, 2500–6000× magnification and large-field low vacuum SED (LED). XRD analysis was performed using a D2 PHASER Instrument, Bruker, Germany.

### 3.4. Adsorption Experiments

The prepared activated carbons were tested for AO7 dye removal, and the highest surface area activated carbon was selected for further evaluation of its AO7 dye removal through batch equilibrium adsorption studies depending on adsorption capacity. Batch adsorption studies were performed using different initial concentrations of AO7 dye prepared by dilution of the feed stock solution of 1000 ppm with distilled water. All adsorption tests were performed in a shaking instrument at ambient temperatures. The pH values (1.5, 3.0, 4.1, 5.1, 7, 9, and 10) of solutions were adjusted before and during the experimental process using 0.1 M HCl or NaOH solutions. The different specified initial solution concentrations were introduced into flasks containing a known amount of the activated carbon at room temperature (24 ± 2 °C). The following parameters were investigated to understand their effects on AO7 dye adsorption onto the activated carbon surface: contact time from 0 to 180 min; pH from 1.5 to 10; adsorbent dosage of 0.75, 1.0, 1.5, 2.0, and 2.5 g L^−1^; and feed solution initial concentration of 100, 150, 200, 300, and 400 mg L^−1^. The initial and equilibrium concentrations were measured utilizing a UV–Vis spectrophotometer at a wavelength of *λ*_max_ 483 nm for AO7 dye using a double-beam spectrophotometer (SPEKOL 1300-ANALYTIK JENA AG-Germany). The suspension was shaken at 200 rpm and at predetermined time intervals. An amount of 1 mL of the clear solution was taken and analyzed using a UV–Vis spectrophotometer. Reported values are averages of the triplicated experiment. AO7 dye removal percentage (%R) was estimated according to the following Equation (4):(4)$$\%R=\frac{C_{i}-C_{e}}{C_{i}}\times 100$$
where *C_i_* and *C_e_* are the AO7 dye concentrations (mg L^−1^) pertaining to initial and equilibrium states of adsorption. Adsorption capacity of AO7 dye after time *t* (min) of adsorption on the activated carbon (*Q_t_*) was calculated using the following Equation (5):(5)$$Q_{t}=\frac{\left(C_{i}-C_{t}\right)}{W}\times V$$
where *C_t_* (mg L^−1^), *V* (L), and *W* (g) are concentration of AO7 dye after time *t*, volume of initial feed solution taken, and the weight of activated carbon used as adsorbent, respectively. The equilibrium adsorption capacity *Q_e_* (mg g^−1^) for the prepared activated carbon was calculated with the following Equation (6):(6)$$Q_{e}=\frac{\left(C_{0}-C_{e}\right)\times V}{W}$$
where the initial dye concentration is *C*_0_ (mg L^−1^) and the dye’s equilibrium concentration is *C_e_* (mg L^−1^). *V* (L) is the dye solution’s volume, while *W* (g) is the adsorbent’s weight. The adsorption kinetic experiments were carried out in the same system with different concentrations of AO7 dye solution (100, 150, 200, 300, and 400 mg L^−1^) using different concentrations of activated carbon (0.75, 1.0, 1.5, 2.0, and 2.5 g L^−1^). In a certain time interval, the remaining concentrations of the AO7 dye were measured with a UV–Vis spectrophotometer.

### 3.5. Isotherm Models Study

Experimental equilibrium values were studied using five isotherm models: Langmuir [87], Freundlich [88], Tempkin [89], Harkins–Jura [90] and Halsey [91]. The best-fitting model was discovered through the use of linear regression. Adsorption occurs uniformly on the active sites of the adsorbent surface, according to Langmuir theory, and there are no molecular interactions between the adsorbate and the adsorbent, which results in the presence of a single-layer deposition on the surface of the adsorbent, which has identical active sites and has the maximum adsorption capacity (*Q_m_*). As soon as the adsorbate occupies an active site, no further adsorption can take place at that site, and there is no further transmigration of the adsorbate in a plane parallel to the surface’s surface tension [69]. The Langmuir isotherm model can be represented in linear relationships as in Equation (7):(7)$$\frac{C_{e}}{Q_{e}}=\frac{1}{Q_{m}\times K_{L}}+\frac{C_{e}}{Q_{m}}$$
where *Q_e_* is the equilibrium adsorption capacity of adsorbent (mg g^−1^), *Q_m_* is the maximum adsorption capacity (mg g^−1^), and *K_L_* is related to the Langmuir adsorption energy constant (L g^−1^*).*

The Freundlich isotherm model [88] assumes the presence of interactions between adsorbed molecules which lead to the formation of multilayer adsorption on the adsorbent surface. Given that Freundlich’s assertion is a linear equation, it implies that when the adsorbate concentration increases, the adsorbate concentration on the adsorbent surface increases as well. It is possible to present Freundlich’s isotherm in linear form using Equation (8):(8)$$\ln\left(Q_{e}\right)=\ln\left(K_{F}\right)+\frac{1}{n}\times ln\left(C_{e}\right)$$
where *K_F_* and *n* represent Freundlich isotherm constants corresponding to adsorption capacity and heterogeneity factor, respectively. A value for 1/*n* below one indicates a normal Langmuir isotherm, while 1/*n* above one is indicative of cooperative adsorption.

According to the Tempkin isotherm model [89], indirect interactions between adsorbate and adsorbate are theorized. This hypothesis stated that, as a result of adsorbent–adsorbate interactions, the heat of adsorption of all molecules in the layer decreases linearly with coverage and that adsorption is characterized by an even distribution of binding energies up to and including a maximum binding energy. Equation (9) explains the linear relation using the Tempkin isotherm model:(9)$$Q_{e}=\frac{RT}{b}\ln\left(K_{T}\right)+\frac{RT}{b}ln\left(C_{e}\right)$$
where *K_T_* (L g^−1^) is the Tempkin isotherm constant, *b* (J mol^−1^) is the adsorption energy (heat of adsorption) variation factor, *R* (8.314 J mol^−1^ K^−1^) is the universal gas constant, and *T* is the absolute temperature in Kelvin. Plotting *Q_e_* versus ln *C_e_* enables determination of isotherm constants *B* (*RT*/*b*) and *K_T_* from slopes and intercepts, respectively. *K_T_* (L/g) is the equilibrium-binding constant corresponding to the maximum binding energy, and constant *B* is related to the heat of the adsorption.

The Harkins–Jura isotherm model [90] takes into account multilayer adsorption as well as the adsorbent’s heterogeneous pore distribution. The Harkins–Jura isotherm is frequently used in the form Equation (10):(10)$$\frac{1}{Q_{e}^{2}}=\left(\frac{B_{HJ}}{A_{HJ}}\right)-\left(\frac{1}{A_{HJ}}\right)log\left(C_{e}\right)$$
where *A_HJ_* and *B_HJ_* are the isotherm constant and multilayer adsorption heterogeneous pore distribution. The Halsey isotherm model [91] is applicable to multilayer adsorption, and heteroporous substances can be used to fit the Halsey equation [91,92]. The Halsey model is frequently used with the following Equation (11):(11)$$\ln\left(Q_{e}\right)=\left[\left(\frac{1}{n_{H}}\right)ln\left(K_{H}\right)\right]+\left(\frac{1}{n_{H}}\right)ln\left(C_{e}\right)$$
where *n_H_* and *K_H_* are Halsey constants.

### 3.6. Adsorption Kinetic Models Study

To comprehend the adsorption kinetics pattern, the collected experimental data were fitted to the described kinetic models. Lagergren’s pseudo-first-order Equation (12) [69,93] is as follows:(12)$$\log\left(Q_{e}-Q_{t}\right)=logQ_{e}-\frac{K_{1}}{2.303}t$$

*K*_1_ (g mg^−1^ min^−1^) is the first-order rate constant, and *Q_t_* and *Q_e_* (mg g^−1^) are the adsorption capacities at time *t* (min) and equilibrium, respectively. Equation (13) can be used to express the pseudo-second-order model [69,94]:(13)$$\frac{t}{Q_{t}}=\frac{1}{K_{2}Q_{e}^{2}}+\frac{t}{Q_{e}}$$
where *K*_2_ is the second-order rate constant (g mg^−1^ min^−1^). The intercept and slope of the plot generated by plotting *t*/*Q_t_* versus *t* can be used to compute *K*_2_ and *Q_e_* [69]. The Elovich kinetic equation is another rate equation that is based on adsorption capacity and is typically represented as Equation (14) [69,95]:(14)$$\frac{dQ_{t}}{dt}=\alpha \;exp\left(-\beta Q_{t}\right)$$
where *α* is the initial adsorption rate (mg g^−1^ min^−1^) and *β* is the desorption constant (g mg^−1^) during the course of a single experiment. It is simplified by assuming *αβt >> t* and by applying the boundary conditions *Q_t_ =* 0 at *t =* 0, and *Q_t_ = Q_t_* at *t = t* simplifies the following Equation (13) to form Equation (15):(15)$$Q_{t}=\frac{1}{\beta }\ln\left(\propto \beta \right)+\frac{1}{\beta }ln\left(t\right)$$

A plot of *Q_t_* versus ln(*t*) should yield a linear relationship with a slope of (1*/β*) and an intercept of (1*/β*) *×* ln(*αβ*). Thus, the constants may be determined from the straight line’s slope and intercept. The experimental data were also fitted using the intraparticle diffusion model Equation (16) [69,96]:(16)$$Q_{t}=K_{dif}t^{0.5}+C$$
where *K*_dif_ (mg g^−1^ min^0.5^) is the intraparticle diffusion rate constant and *C* (mg g^−1^) is a constant revealing the thickness of the boundary layer. The liquid film diffusion model [69,97] can be used when the transfer of solute molecules from the liquid phase to the solid phase boundary plays a major role in adsorption (Equation (17)):(17)$$\ln\left(1-F\right)=K_{FD}\left(t\right)$$
where *K_FD_*
*and F* (*F = Q_t_*/*Q_e_*) are the film diffusion rate constant and the fractional attainment of equilibrium, respectively.

### 3.7. Optimization Study Response Surface Methodology (RSM)

In contrast to the other forms of RSM designs, such as central composite or Box–Behnken, the historical data design and optimal custom design of RSM enable the construction of mathematical models based on previously collected experimental data [98]. The D-optimal design was studied using the Stat-Ease Design-Expert v13.0.5.0 program to investigate the adsorption of AO7 dye from the adsorbent coating. Using response surface methods (RSM), the impacts of three independent variables (A: contact time, B: activated carbon dose, and C: beginning AO7 dye concentration) on the response (*R*: dye removal %) were investigated. RSM is a statistical technique that utilizes quantitative data from appropriate experiments to establish the regression model equations and operating parameters. The optimization procedure consists of three steps: executing a statistically planned experiment, determining mathematical model coefficients and response value prediction, and checking the produced model’s suitability [83]. The experiment’s range and variable are listed in Table 8.

Six axial points, eight factorial points, and six replicates at the central point were used to create the best bespoke design for the three independent variables. The factors chosen were varied on a five-level scale (−α, −1, 0, 1, +α). The number of experiment runs was calculated based on Equation (18):(18)$$N=2^{k}+2k+C=2^{3}+2.3+6$$
where *N* is the total number of runs, *k* denotes the total number of factors to be tested, and *C* denotes the total number of experiments completed at the center. Table 8 shows the lower and upper bounds for each factor. State-Ease Design-Expert v 13.0.5.0 was used to generate the experiment data matrix. Study of variance was used to conduct a statistical analysis of the generated model (ANOVA). Surface contour plots were used to investigate the relationships between factors.

## 4. Conclusions

The synthesis of activated carbon from *Pisum sativum* pea pods was demonstrated in this work employing a 2:1 ratio of ZnCl_2_ and a 10 percent solution of KOH as activation reagents. The impregnated biomass was activated at different temperatures (600, 700, and 800 °C) under N_2_ and CO_2_ gases for 60 min. It is worth noting that the chemical activation reagents and the gas medium to be treated had a significant effect on the surface area and pore structure of activated carbons, with ZnCl_2_ and CO_2_ being preferred. To optimize the activation settings, the response surface methodology was successfully employed. The optimal conditions for the removal of AO7 dye were determined to be carbon impregnated with a 2:1 ratio of ZnCl_2_ under CO_2_ gas at 800 °C and AO7 dye solution at pH 1.5, which achieved a removal percentage of 100%. Using SEM images, it was discovered that a pore structure had formed in the AC-ZnCl_2_ during the activation phase. The AC-ZnCl_2_ under CO_2_ at 800 °C exhibited a high surface area (*S*_BET_) of 1300 to 1500 m^2^ g^−1^, a total pore volume of 0.6176 cm^3^ g^−1^, and a mean pore diameter of 1.901 nm. Based on the experimental maximum adsorption capacity at equilibrium was 467.16 mg g^−1^ and the maximum percentages removal of AO7 dye was ~100%. It is possible to conclude that the adsorption mechanism of AO7 dye onto AC-ZnCl_2_ is a process of monolayer formation and governed by physisorption, which is best modelled with the Langmuir isotherm and obeys the pseudo-second-order and film diffusion kinetic models. The maximum adsorption capacity (*Q_m_*) for AO7 dye determined in this work using a selective activation approach was 473.93 mg g^−1^, which is significantly higher than the adsorption capacity of numerous activated carbons previously reported in the literature. As a result, it has the potential to be employed as a viable adsorbent biomaterial for the removal of AO7 dye from aqueous solutions.

## Data Availability

Not applicable.

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
