# Peer review of "Microporous Activated Carbon from Pisum sativum Pods Using Various Activation Methods and Tested for Adsorption of Acid Orange 7 Dye from Water"

_molecules, 2022, doi:10.3390/molecules27154840_

Round 1

Reviewer 1 Report

Comments from Reviewer
Title: Microporous activated carbon from Pisum sativum pods using various activation methods and tested for adsorption of Acid Orange 7 dye from water
The current form's presentation of methods and scientific results is unsatisfactory for publication in the Molecules journal. The minor and significant drawbacks to be addressed can be specified as follows:
1.    Abstract, 48.88 and 69.95. Too high accuracy. See also 473.93.
2.    Abstract. (i) AO7 ---> Acid orange 7 (ii) response factor model (RSM) ---> response factor model.
3.    Page 4, Scheme 1. Unbalanced reaction equation: KOH = K2O + H2O.
4.    Page 4, Schemes 1 and 2. Why reactants are placed above the arrows.
5.    The article is too long and contains unnecessary data. The number of results does not necessarily affect the quality of the article. See Tab. 1, Figs. 7 – 9, Tabs. 3 and 4, Tabs. 6 and 7. Most of this data can be moved to Supporting Information.
6.    Tab. 1. (i) red fonts? (ii) most of this data should be moved to Supporting Information. (iii) Vm, mean pore diemeter, and total pore volume cannot be equated with the BET method. Vm, mean pore diameter, and total pore volume - how were they calculated? Using the BET method? (iv) Data for t-plot, MP, and BJH des are not needed. (v) Cm3 ---> cm3 (vi) Vhy double notation? as, BET (m2 ∕g) or SBET (m2 ∕g)???? Vm (cm3/g) or Vm (cm3/g)??? (vii) Too high accuracy of the collected values.
7.    Fig. 1. (i) Coupled Two Theta/Theta? (ii) WL?
8.    Fig. 4. Scale bars and numerical values indicating resolution are poorly visible.
9.    Figs. 1 and 5. The marking of the samples is difficult to understand. Please compare the content of the legends.
10.    Fig. 6(b). Why do the adsorption and desorption branches not come together?
11.    Fig. 6, figure captions. Adsorption Desorption ---> Adsorption/desorption.
12.    Figs. 7 - 9. Showing this data is unnecessary. These are standard procedures. These curves will not interest readers. Everyone ignores them.
13.    Fig. 10. (i) why the authors use the BJH method for this type of material. They should use the NLDFT method. The tested materials are micro-mesoporous systems. The NLDFT method is now regarded as a standard tool for examining the structure of this type of material. (ii) rp – radius? diameter? (iii) what value of the pore width is detected by the BJH method? (iv) why can't this method be used for micropores?
14.    Fig. 11. how many times were the measurements repeated? Why haven't measurements been made at pH = 6? is the value for pH = 5 incorrect?
15.    The values of the adjusted coefficient of determination should be considered. Classical R2 shows how well terms (data points) fit a curve or line. Adjusted R2 also indicates how well paraterms fit a curve or line but adjust for the number of paraterms in a model. The comparison of Adjusted R2 allows only to compare different models (with different variables). The next problem, the results obtained indicate compliance with the rule - the more best fit parameters, the better the compatibility between experimental and theoretical data.
16.    Tab. 2. (i) why analyze so many models. (ii) Harkins-Jura isotherm: AHJ = 50000??? -= 10000? = 5000? Where do these values come from?
17.    Tabs. 3 and 4. These tables should be collected in Supp. Inf.
18.    2.3.5 Adsorption mechanism. How this mechanism differs from others? It was clarified a long time ago.
19.    Page 33. main pore size? mean!!!!
20.    (i) Literature should also be standardized: the size of letters in the titles of journals, initials of names, and the size of letters in the titles of articles. (ii) too many refernces. (iii) 28 self-citations [4-6, 10, 12, 14-16, 18-25, 29, 44, 49, 55-58, 72, 73, 76-78] on 107 references 26% (not acceptable).

Sincerely,
The reviewer.

Author Response

Response to reviewer's comments

Manuscript ID: molecules-1819507

Title: Microporous activated carbon from Pisum sativum pods using various activation methods and tested for adsorption of Acid Orange 7 dye from water

Authors: Mohamed A. El-Nemr, Ahmed El Nemr *, Mohamed A. Hassaan, Safaa
Ragab, Luigi Tedone, Giuseppe De Mastro, Antonio Pantaleo

Dear Dr. Ana-Maria Sacara,
Section Managing Editor, MDPI Romania

I would like to submit our response to the comments raised by referee about our manuscript ID: molecules-1819507.

Reviewer #1

Comment #1

  1. Abstract, 48.88 and 69.95. Too high accuracy. See also 473.93.

Response: Thank you for comment.

These are the outcomes of the SEM image analysis and Qm from the LM, which were reported as we measured. We were unable to alter the outcomes of the experiment or analysis.

Comment #2

  1. Abstract. (i) AO7 ---> Acid orange 7 (ii) response factor model (RSM) ---> response factor model.

Response: Thank you for comment.

Corrected.

Comment #3

  1. Page 4, Scheme 1. Unbalanced reaction equation: KOH = K2O + H2O.

Response: Thank you for comment.

Corrected.

Comment #4

  1. Page 4, Schemes 1 and 2. Why reactants are placed above the arrows.

Response: Thank you for comment.

Corrected.

Comment #5

  1. The article is too long and contains unnecessary data. The number of results does not necessarily affect the quality of the article. See Tab. 1, Figs. 7 – 9, Tabs. 3 and 4, Tabs. 6 and 7. Most of this data can be moved to Supporting Information.

Response: Thank you for comment.

I like the manuscript to have all the information because if the reader is using the supporting information he is usually needs to read the paper into two files which is not suitable for the reader.

Comment #6

  1. Tab. 1. (i) red fonts? (ii) most of this data should be moved to Supporting Information. (iii) Vm, mean pore diemeter, and total pore volume cannot be equated with the BET method. Vm, mean pore diameter, and total pore volume - how were they calculated? Using the BET method? (iv) Data for t-plot, MP, and BJH des are not needed. (v) Cm3 ---> cm3 (vi) Vhy double notation? as, BET (m2 ∕g) or SBET (m2 ∕g)???? Vm (cm3/g) or Vm (cm3/g)??? (vii) Too high accuracy of the collected values.

Response: Thank you for comment.

All these data were obtained from the isotherm analyses of the prepared activated carbon. Some of these data obtained directly from the models and the other can be calculated following the instrument catalog. Please see below the image of BET analysis as an example. Since the pour is between micro and measo pour diameter we should use different models for analysis of the isotherm curve. Please see also the references for the analysis in the references list.

Comment #7

  1. Fig. 1. (i) Coupled Two Theta/Theta? (ii) WL?

Response: Thank you for comment.

This figure was reported as obtained from the instrument analysis.

Comment #8

  1. Fig. 4. Scale bars and numerical values indicating resolution are poorly visible.

Response: Thank you for comment.

The scale bars and numerical values are not possible to maximize more therefore I added to the figure legend.

Comment #9

  1. Figs. 1 and 5. The marking of the samples is difficult to understand. Please compare the content of the legends.

Response: Thank you for comment.

The figure legends were corrected.

Comment #10

  1. Fig. 6(b). Why do the adsorption and desorption branches not come together?

Response: Thank you for comment.

This depends on the type of carbons which makes the types of isotherm curves.

Comment #11

  1. Fig. 6, figure captions. Adsorption Desorption ---> Adsorption/desorption.

Response: Thank you for comment.

Corrected.

Comment #12

  1. Figs. 7 - 9. Showing this data is unnecessary. These are standard procedures. These curves will not interest readers. Everyone ignores them.

Response: Thank you for comment.

I like the manuscript to have all the information because if the reader is using the supporting information he is usually needs to read the paper into two files which is not suitable for the reader.

Comment #13

  1. Fig. 10. (i) why the authors use the BJH method for this type of material. They should use the NLDFT method. The tested materials are micro-mesoporous systems. The NLDFT method is now regarded as a standard tool for examining the structure of this type of material. (ii) rp – radius? diameter? (iii) what value of the pore width is detected by the BJH method? (iv) why can't this method be used for micropores?

Response: Thank you for comment.

Not available for us now. We will take this in my consideration in the future work.

Comment #14

  1. Fig. 11. how many times were the measurements repeated? Why haven't measurements been made at pH = 6? is the value for pH = 5 incorrect?

Response: Thank you for comment.

All the experimental work was repeated three times and only the mean value was reported. No need to make pH 6 since the best pH was 1.5.

Comment #15

  1. The values of the adjusted coefficient of determination should be considered. Classical R2 shows how well terms (data points) fit a curve or line. Adjusted R2 also indicates how well paraterms fit a curve or line but adjust for the number of paraterms in a model. The comparison of Adjusted R2 allows only to compare different models (with different variables). The next problem, the results obtained indicate compliance with the rule - the more best fit parameters, the better the compatibility between experimental and theoretical data.

Response: Thank you for comment.

In most of our previous work, we demonstrated that when the model becomes applicable to empirical data, the computed qe must be equal to the empirical qe. Adjusted R2 is not applicable at this point.  

Comment #16

  1. Tab. 2. (i) why analyze so many models. (ii) Harkins-Jura isotherm: AHJ = 50000??? -= 10000? = 5000? Where do these values come from?

Response: Thank you for comment.

It may be due to the lower R2 value which may indicates that the model is not abdicable to the experimental data.

Comment #17

  1. Tabs. 3 and 4. These tables should be collected in Supp. Inf.

Response: Thank you for comment.

I like the manuscript to have all the information because if the reader is using the supporting information he is usually needs to read the paper into two files which is not suitable for the reader.

Comment #18

  1. 2.3.5 Adsorption mechanism. How this mechanism differs from others? It was clarified a long time ago.

Response: Thank you for comment.

This is merely to illustrate the type of adsorption mechanism used in prepared AC.

Comment #19

  1. Page 33. main pore size? mean!!!!

Response: Thank you for comment.

Corrected.

Comment #20

  1. (i) Literature should also be standardized: the size of letters in the titles of journals, initials of names, and the size of letters in the titles of articles. (ii) too many refernces. (iii) 28 self-citations [4-6, 10, 12, 14-16, 18-25, 29, 44, 49, 55-58, 72, 73, 76-78] on 107 references 26% (not acceptable).

Response: Thank you for comment.

Some of the self-citations were removed.

Reviewer #2

The study was well designed and conducted and is relevant for researchers and developers of adsorption technologies. Therefore, I advise accepting it for publication after some corrections:

Response: Thank you for comment.

Authors should check and correct the spelling in the following excerpts:

Comment #1

3.3 “The samples were pre-treated at 300 μ under flow of nitrogen gas.”

Response: Thank you for comment.

Corrected to be "300 °C"

Comment #2

3.4 “The initial and equilibrium concentrations were measured using a UV-Vis spectrophotometer at wavelength of max 483 nm for AO7 dye using a double beam spectrophotometer…”

Response: Thank you for comment.

Corrected to be "λmax 483 nm"

Comment #3

3.4 In describing the equations, the authors use italics at times and not at others. I suggest standardizing the writing of all variables in the text to italics.

Response: Thank you for comment.

Corrected.

Comment #3

In Figure 5, I suggest not using axis color similar to curves from a sample. Authors could differentiate TGA and DTA, using solid lines for one and dotted lines for the other.

Response: Thank you for comment.

The shape of the TGA and DTA curve is well known and we use one color for each type of material. Any change may increase the difficulty.

Comment #4

Standardize the formatting of figures: strokes and fonts

Response: Thank you for comment.

We attempted to make the figures as accurate as possible.

Comment #4

Table 7 can be removed from the main text and inserted in the supplementary information.

Response: Thank you for comment.

I like the manuscript to have all the information because if the reader is using the supporting information he is usually needs to read the paper into two files which is not suitable for the reader.

The authors present the linearized and non-linearized forms of isothermal equations. This is already well established in the literature, choose only one form of representation.

Response: Thank you for comment.

Corrected.

Reviewer #3

Comments and Suggestions for Authors

Major revisions:

Comment #1

  1. The novelty of this research should be inserted in the text clearly.

Response: Thank you for comment.

Inserted before the introduction.

Comment #2

  1. The advantages and disadvantages of the synthesized adsorbent should be investigated.

Response: Thank you for comment.

The advanced and disadvantages of the prepared AC were reported through the manuscript.

Comment #3

  1. The stability of the synthesized adsorbent should be presented by XRD.

Response: Thank you for comment.

The stability of AC is well known.

Comment #4

  1. Dye adsorption thermodynamic should be presented.

Response: Thank you for comment.

The amount of the presented work is too much and addition of more work is to difficult now.

Comment #5

  1. The “introduction” and “results and discussion” sections of the manuscript can be strengthened and supported with some papers related to the literature and cited (optional for authors):

Journal of cleaner production 222 (2019), 669-684;

Journal of Molecular Liquids 269 (2018), 217-228;

Journal of Cleaner Production 183 (2018), 1197-1206;

Response: Thank you for comment.

Two of the proposed literature were used and inserted into the introduction text and references list.

I would like to thank the referees and editor for their valuable advice and comments which improved our work.

Best regards

Ahmed El Nemr

Reviewer 2 Report

In this study, the authors produced different methods and characterized activated carbon from Pisum sativum pods, which was applied in the adsorption of Acid Orange 7 dye. The study was well designed and conducted and is relevant for researchers and developers of adsorption technologies. Therefore, I advise accepting it for publication after some corrections:

Authors should check and correct the spelling in the following excerpts:

3.3 “The samples were pre-treated at 300 μ under flow of nitrogen gas.”

3.4 “The initial and equilibrium concentrations were measured using a UV-Vis spectrophotometer at wavelength of max 483 nm for AO7 dye using a double beam spectrophotometer…”

3.4 In describing the equations, the authors use italics at times and not at others. I suggest standardizing the writing of all variables in the text to italics.

In Figure 5, I suggest not using axis color similar to curves from a sample. Authors could differentiate TGA and DTA, using solid lines for one and dotted lines for the other.

Standardize the formatting of figures: strokes and fonts

Table 7 can be removed from the main text and inserted in the supplementary information.

The authors present the linearized and non-linearized forms of isothermal equations. This is already well established in the literature, choose only one form of representation.

Author Response

(The authors gave the same response as above.)

Reviewer 3 Report

Comments:

Major revisions:

1. The novelty of this research should be inserted in the text clearly.

2. The advantages and disadvantages of the synthesized adsorbent should be investigated.

3. The stability of the synthesized adsorbent should be presented by XRD.

4. Dye adsorption thermodynamic should be presented.

5. The “introduction” and “results and discussion” sections of the manuscript can be strengthened and supported with some papers related to the literature and cited (optional for authors):

Journal of cleaner production 222 (2019), 669-684;

Journal of Molecular Liquids 269 (2018), 217-228;

Journal of Cleaner Production 183 (2018), 1197-1206;

Author Response

(The authors gave the same response as above.)

Round 2

Reviewer 1 Report

Comments from Reviewer

Title: Microporous activated carbon from Pisum sativum pods using various activation methods and tested for adsorption of Acid Orange 7 dye from water

The current form's presentation of methods and scientific results is still unsatisfactory for publication in the Molecules journal. Reading the revised manuscript, it seems that some comments have been completely ignored.

1. Tab. 1. Red fonts? Why? Please give the respective information in the table captions? Such a presentation of the results is confusing to the readers!

2. Fig. 1. Coupled Two Theta/Theta? (ii) WL? Please explain it!!! Not everyone is a specialist. That's what the figure captions under the plots explain these kinds of things. Different instruments have different software, and not every reader needs to know what this text mean.

3. Fig. 1. Unfortunately, apart from the legend, the sample designations (Peas_Raw_KOH, AC KOH_CO2_800, Raw Pea pods-ZnCl2, and AC_ZnCl2_CO2_800) are not used anywhere in the text. Please standardize this and use the same marking everywhere. In general, the authors are pretty comfortable with naming samples - see the legend in Figure 5. I believe this is the result of using print screens. All programs allow you to export data in the form of dat/txt files in Ascii code and prepare the new figures using specialized software for making drawings/figures - for example, excell, grapher, sigmaplot or origin.

4. Fig. 6(b). Why do the adsorption and desorption branches not come together?
Response: Thank you for comment.
This depends on the type of carbons which makes the types of isotherm curves.
My response. Why are these differences observed? What happens with the material that the ACs does not swell under the influence of ZnCl2, and so does it under the influence of CO2? Is swelling permanent? Is the material destroyed during N2 adsorption for AC_KOH samples? Did the authors check this in detail - for example, by desorbing the samples after the first nitrogen measurement and re-adsorbing the nitrogen, and considering the second cycle (nitrogen re-adsorption)? This behaviour of the material is very important and crucial from the point of view of adsorption. This raises the question of whether a similar behavior of the material can be related to adsorption from the liquid phase.

5. Figs. 7 - 9. Showing this data is unnecessary. These are standard procedures. These curves will not interest readers. Everyone ignores them.
Response: Thank you for comment. I like the manuscript to have all the information because if the reader is using the supporting information he is usually needs to read the paper into two files which is not suitable for the reader.
My response: No one publishes drawings showing how to get the apparent BET values !!! The number of figures does not prove the quality of the article. I have reviewed about 200 articles for MDPI publishers, and no one has published such data before. If the authors absolutely want to publish these results, I again suggest moving them to supporting information.

Sincerely,
    The reviewer.

Author Response

Top of Form

Comments and Suggestions for Authors

Comments from Reviewer

Title: Microporous activated carbon from Pisum sativum pods using various activation methods and tested for adsorption of Acid Orange 7 dye from water

The current form's presentation of methods and scientific results is still unsatisfactory for publication in the Molecules journal. Reading the revised manuscript, it seems that some comments have been completely ignored.

Response:

Sorry for this mistake and we didn't mean to ignore it and we have replied to all of them in the below lines.

Comment #1

  1. Tab. 1. Red fonts? Why? Please give the respective information in the table captions? Such a presentation of the results is confusing to the readers! 

Response:

The red color was changed to normal black color.

Comment #2

  1. Fig. 1. Coupled Two Theta/Theta? (ii) WL? Please explain it!!! Not everyone is a specialist. That's what the figure captions under the plots explain these kinds of things. Different instruments have different software, and not every reader needs to know what this text mean.

Response:

We have corrected it to be 2Theta (degree)

Moreover, these symbols are familiar in XRD figures and is used in this technique

Where, θ is the angle between the incident beam and the crystallographic reflecting plane. It is also equal to the angle between the reflected beam and the crystallographic plane.

In powder x-ray diffractometry the powder sample is loaded in a small disc-like container and its surface is carefully flattened. The disc is put on one axis of the diffractometer and tilted by an angle θ while a detector, a scintillation counter, rotate around it on an arm at twice this angle. This configuration is known under the name Bragg–Brentano θ-2θ. Another configuration is the Bragg–Brentano θ-θ configuration in which the sample is stationary while the X-ray tube and the detector are rotated around it.

The angle formed between the x-ray source and the detector is 2θ. This configuration is most convenient for loose powders. Thus the 2 θ is the angle between transmitted beam and reflected beam. In an experiment, the transmitted and the reflected beam can be observed, but the crystallographic plane cannot be observed, so 2 θ is an experimentally measurable quantity. That's why in the x-ray powder diffractometry analysis, we use intensity vs. 2 θ plots.

For WL or λ:

Scherrer’s equation to calculate the nano crystallite size (L):

 L=Kλ/β.cosθ

to calculate the nano crystallite size (L) by XRD radiation of wavelength λ (nm)  (λ  = 1.54060 Å (in the case of CuKa1), from measuring full width at half maximum of peaks (β) in radian located at any 2θ in the pattern and K is constant and is usually taken as about 0.89 or 0.9.

Comment #3

  1. Fig. 1. Unfortunately, apart from the legend, the sample designations (Peas_Raw_KOH, AC KOH_CO2_800, Raw Pea pods-ZnCl2, and AC_ZnCl2_CO2_800) are not used anywhere in the text. Please standardize this and use the same marking everywhere. In general, the authors are pretty comfortable with naming samples - see the legend in Figure 5. I believe this is the result of using print screens. All programs allow you to export data in the form of dat/txt files in Ascii code and prepare the new figures using specialized software for making drawings/figures - for example, excell, grapher, sigmaplot or origin.

Response:

Corrected. Thank you.

Comment #4

  1. Fig. 6(b). Why do the adsorption and desorption branches not come together?

Response: Thank you for comment.

This depends on the type of carbons which makes the types of isotherm curves.
My response. Why are these differences observed? What happens with the material that the ACs does not swell under the influence of ZnCl2, and so does it under the influence of CO2? Is swelling permanent? Is the material destroyed during N2 adsorption for AC_KOH samples? Did the authors check this in detail - for example, by desorbing the samples after the first nitrogen measurement and re-adsorbing the nitrogen, and considering the second cycle (nitrogen re-adsorption)? This behaviour of the material is very important and crucial from the point of view of adsorption. This raises the question of whether a similar behavior of the material can be related to adsorption from the liquid phase.

Response:

We do not have an explanation to behavior. We will think about this observation in the next research work. Thank you very much.

Comment #4

  1. Figs. 7 - 9. Showing this data is unnecessary. These are standard procedures. These curves will not interest readers. Everyone ignores them. 
    Response: Thank you for comment. I like the manuscript to have all the information because if the reader is using the supporting information he is usually needs to read the paper into two files which is not suitable for the reader.

My response: No one publishes drawings showing how to get the apparent BET values !!! The number of figures does not prove the quality of the article. I have reviewed about 200 articles for MDPI publishers, and no one has published such data before. If the authors absolutely want to publish these results, I again suggest moving them to supporting information.

Response:

Thank you for your comment and sorry for this inadvertent error

We have moved them to the supplementary and supporting materials AS 1S, 2S and 3S and we highlighted it in yellow color.

All figures were renumbered again after moving figs. 7-9 to the Supp. Materials. And we have highlighted them.

I would like to thank the referees and editor for their valuable advice and comments which improved our work.

Best regards

Ahmed El Nemr

Bottom of Form

Reviewer 3 Report

Accept

Author Response

Thank you very much.